# Covalent Organic Frameworks-TpPa-1 as an Emerging Platform for Electrochemical Sensing

**DOI:** 10.3390/nano12172953

**Published:** 2022-08-26

**Authors:** Gang Li, Baiqing Yuan, Sidi Chen, Liju Gan, Chunying Xu

**Affiliations:** School of Chemistry and Materials Science, Ludong University, Yantai 264025, China

**Keywords:** covalent organic frameworks, electrochemical sensor, hydrazine, nitrophenol, nitrogen doped carbon, reduced glutathione

## Abstract

Covalent organic frameworks (COFs) are a new type of metal-free porous architecture with a well-designed pore structure and high stability. Here an efficient electrochemical sensing platform was demonstrated based on COFs TpPa-1 constructed by 1,3,5-triformylphloroglucinol (Tp) with p-phenylenediamine (Pa-1), which possesses abundant nitrogen and oxo-functionalities. COFs TpPa-1 exhibited good water dispersibility and strong adsorption affinities for Pd^2+^ and thus was used as loading support to modify Pd^2+^. The Pd^2+^-modified COFs TpPa-1 electrode (Pd^2+^/COFs) showed high electrocatalytic activity for both hydrazine oxidation reaction and nitrophenol reduction reaction. In addition, TpPa-1-derived nitrogen-doped carbon presented high activity for the electro-oxidation of reduced glutathione (GSH), and sensitive electrochemical detection of GSH was achieved. The presented COFs TpPa-1 can be utilized as a precursor as well as support for anchoring electro-active molecules and nanoparticles, which will be useful for electrochemical sensing and electrocatalysis.

## 1. Introduction

The synthesis and construction of efficient electrode materials to gain high sensitivity and selectivity is crucial for electroanalysis. The successful development of nonenzymatic electrochemical sensors depends critically on electrode materials with excellent electrocatalytic properties because many analytes are electrochemically inactive when using traditional electrodes. Up to now, different categories of materials, including noble metals, metal oxides, and carbons (such as CNT and graphene), have been applied in electrochemical sensing [1]. Recently, porous materials have demonstrated advantages in constructing electrochemical sensors due to their high specific surface areas and powerful adsorption and loading capability [2].

Metal–organic frameworks (MOFs) and covalent organic frameworks (COFs), as two emerging families of crystalline porous materials, have been explored in electroanalysis and electrocatalysis [3,4]. Compared to conventional porous materials such as zeolites, porous oxides, and carbons, MOFs and COFs show some intriguing features for electrochemical applications, including inter-connected porosity, tunable intra-framework chemical functionality, and high surface area [5]. Wang et al. synthesized graphene aerogel (GA) and metal–organic framework (MOF) composites via the in situ growth of the MOF UiO-66-NH_2_ crystal on the GA matrix. The interface showed high sensitivity and selectivity for the electrochemical sensing of multiple heavy-metal ions. GA was employed as the backbone for UiO-66-NH_2_, which also improved the electrical conductivity of the composites by accelerating the electron transfer in the matrix. UiO-66-NH_2_ provided a binding site for heavy-metal ions due to the interaction between hydrophilic species and metal cations [6]. Xu et al. developed a facile furazolidone sensor based on COFs [7]. In this work, 1,3,5-tris-(4-amino-phenyl) benzene (TAPB) and terephthaldicarboxaldehyde (TPA) were used as monomers to synthesize the COF, which was then self-assembled on the surface of the NH_2_-CNTs via a simple and rapid one-pot strategy at room temperature. The prepared COF@NH_2_-CNT composites exhibited a high surface area and excellent conductivity, thus achieving high sensitive detection of furazolidone. However, most MOFs are not chemically stable in water. If the MOF is unstable in the aqueous buffer used for electroanalysis, the MOF coating modified on the electrode surface may suffer from dissolving or converting into MOF-derived metal hydroxides or oxides [8]. Unlike MOFs, COFs are regularly integrated into metal-free crystalline organic structures through strong covalent bonds between the organic building blocks, leading to more robust frameworks and relatively lower densities. Many COFs possess the sp^2^ carbon networks with a high degree of p-conjunction and thus lead to at least a modest electronic conductivity, which facilitates the electron transfer rate. Furthermore, the presence of electronegative atoms (N and O) and π bonds within COFs can allow more specific target analyte recognition through the formation of hydrogen bonds or π-π bonds, thus improving the sensitivity and selectivity of COF-based electrochemical sensors [7]. These unique characteristics of COFs make them promising materials for electrochemical sensing.

COFs-derived porous carbons are also considered ideal platforms for electrochemical sensing due to their high conductivity, porosity, chemical and thermal stability, high electrocatalytic activity, and powerful supporting performances for anchoring molecules and nanomaterials. Most intrinsic electrocatalytic activities are contributed to the oxo-functionalities and edge-plane-like sites (defects) present on the surface of carbon materials [9]. COFs are often rich with nonmetal elements such as N, B, S, and P, which can induce heteroatom doping into derived carbons. Heteroatom doping forms charge delocalization and changes in the electronic structure due to the difference in atomic size, bond length, and coordination between the carbon atoms and dopants, leading to abundant defect sites [10]. N-doping can increase the surface polarity and wettability of the electrodes and also improve the electrocatalytic activity and adsorption [11,12]. Here COFs TpPa-1 constructed by 1,3,5-triformylphloroglucinol (Tp) with p-phenylenediamine (Pa-1) possess abundant nitrogen and oxo-functionalities, which was demonstrated for Pd^2+^ immobilization and fabricating nitrogen-doped porous carbon. The presented electrocatalysts showed high responses for hydrazine oxidation, nitrophenol reduction, and glutathione (GSH) oxidation.

## 2. Experimental

### 2.1. Chemicals and Solutions

GSH, palladium chloride (PdCl_2_), hydrazine (wt. 80%), graphite powder, liquid paraffin, p-nitrophenol (PNP), and o-nitrophenol (ONP) were purchased from Sigma-Aldrich (St. Louis, MO, USA). COFs-TpPa-1 was purchased from Nanjing XFNano Materials Tech Co., Ltd. (Nanjing, China) The pore size of COFs-TpPa-1 is in the range of 1.5–1.8 nm with a BET specific surface area of ~1360 m^2^/g. All of the other chemicals were of analytical reagent grade and used without further purification.

### 2.2. Apparatus

The morphologies were characterized by scanning electron microscopy (SEM) (Hitachi SU8010, Tokyo, Japan) and transmission electron microscopy (TEM) (JEM-2100, JEOL, Tokyo, Japan). The X-ray photoelectron spectroscopy (XPS) was recorded on a Thermo ESCALAB 250 Xi spectrometer (Waltham, MA, USA) fitted with a monochromatic Al Kα X-ray source. X-ray diffraction patterns were collected by a Bruker D8 (Billerica, MA, USA) advance powder X-ray Cu Ka radiation diffractometer. All of the electrochemical experiments were conducted on a CHI 750E electrochemical workstation with a conventional three-electrode system consisting of a modified working electrode, platinum coil auxiliary electrode, and Ag/AgCl (saturated KCl) reference electrode. Electrochemical impedance spectra (EIS) were carried out in 0.1 M KCl containing 5.0 mM Fe (CN)_6_^3−^/^4−^ in the frequency range of 1 MHz to 0.1 Hz at 0.25 V.

### 2.3. Electrode Preparation and Modification

#### 2.3.1. Preparation of Pd^2+^/COFs/GCE

Prior to modification, the glassy carbon electrode (GCE) (3 mm diameter, 0.07 cm^2^) was successively polished with 1, 0.3, and 0.05 µm alumina paste to a mirror finish and then rinsed with deionized water followed by ultrasonic treatment in water and ethanol, respectively. The COFs-modified GCE (COFs/GCE) was prepared by dropping 10.0 µL COFs aqueous suspension (2.0 mg/mL) on a cleaned GCE and dried at room temperature. In order to anchor Pd^2+^, the prepared COFs/GCE was dipped in 10 mM PdCl_2_ solution for 30 min, which was then washed with water and left to dry at room temperature (Pd^2+^/COFs/GCE). Pd^2+^/GCE was prepared by soaking the GCE in Pd^2+^ solution by electroless deposition. The electroactive surface areas of the different electrodes could be calculated by the Randles–Sevcik equation using the CV responses of the corresponding electrodes in 0.1 M KCl solution in the presence of 5.0 mM K_3_Fe (CN)_6_ at different scan rates at 298 K (Appendix A).

#### 2.3.2. Preparation of PCs/CPE

The nitrogen-doped porous carbon materials (PCs) were prepared by calcination. A number of COFs were calcined at 800 °C (heating rate of 10 °C min^−1^) for 80 min under nitrogen flow in the tubular furnace, and black carbonized materials were obtained after cooling to room temperature naturally. To prepare the PCs modified electrode, 3.0 mg PCs were ultrasonically dispersed in 1.0 mL water for 1 h. The carbon paste electrode (CPE) was prepared by thoroughly mixing 1 mg of the graphite powder and 300 µL of liquid paraffin using an agate mortar and pestle for 40 min. Then, a suitable amount of the above mixture was filled firmly into one glass tube. The electrical contact was made by inserting a copper wire into the paste in the inner hole of the tube. Finally, 5.0 µL of 3.0 mg/mL PC suspension was dropped onto the freshly prepared CPE surface and dried at room temperature (PCs/CPE).

## 3. Results and Discussion

### 3.1. Electrochemical Sensing of Hydrazine

#### 3.1.1. Characterization of Pd^2+^/COFs/GCE

The surface morphology and elemental composition of Pd^2+^/COFs/GCE were investigated by SEM and XPS. As shown in Figure 1, the SEM image of Pd^2+^/COFs/GCE shows the rough and inter-connected porous surface. The morphology is mainly composed of complicated long strips, indicating that COFs were modified on the GCE surface. XPS was also carried out to explore the surface composition and chemical states (Figure 2). The characteristic peaks of C, N, O, and Pd elements are found in the XPS survey spectrum (Appendix A). The O1s XPS spectrum exhibits four peaks centered at 530.79, 531.96, 533.18, and 535.29 eV (Figure 2a), which can be attributed to C−OH, C=O, C−O, or H−O, and chemisorbed oxygen in carboxylic groups or water (H-O-H), respectively [13]. The abundant oxo functionalities present on COFs can improve their dispersion in water, which is beneficial for the modification of COFs on the electrode surface. The N1s XPS spectra show two characteristic peaks at 400.06 and 403.53 eV (Figure 2c), which are assigned to the C=N and C-N groups, respectively [14]. Figure 2b displays the XPS Pd3d spectra of Pd^2+^/COFs, including two peaks at 338.10 and 343.40 eV, corresponding to the peak of the low energy band (Pd3d 3/2) and high energy band (Pd3d 5/2), respectively. The Pd^2+^ species might be attached to COFs via N coordination [15]. The results confirm the existence of Pd on the composite surface, and the Pd^2+^ is successfully loaded on the COFs. The XPS data showed that the atomic amount of Pd is 0.16%. In order to investigate the stability of COFs, the XRD patterns of COFs, Pd^2+^/COFs, and Pd^2+^/COFs soaked in 0.1 M NaOH were performed (Figure 3). The characteristic peaks of these samples observed in XRD patterns were in accord with the reported literature [15,16] and remained unchanged, indicating that Pd^2+^ modification and NaOH solution soaking did not change the crystalline structure. The results showed that COF-TpPa-1 possessed good chemical stability. It was also found that COF-TpPa-1 was still stable even though soaked in 1.0 mol/L HNO_3_, 6.0 mol/L NaOH, and HCl for three days [16].

#### 3.1.2. Electrochemical Oxidation of Hydrazine

Figure 4 shows the CVs of COFs/GCE, Pd^2+^/GCE, and Pd^2+^/COFs/GCE in 0.1 M NaOH in the absence and presence of 5.0 mM hydrazine at a scan rate of 100 mV s^−1^. At COFs/GCE, the hydrazine oxidation began at the potential of 0.25 V (Figure 4a) and reached the oxidation peak at about 0.75 V. However, for Pd^2+^/GCE, two oxidation peaks were observed at −0.25 and 0.75 V, respectively (Figure 4b). Pd^2+^ was reduced to Pd metal by electroless deposition when GCE was immersed in a Pd^2+^ solution [17]. The first oxidation wave was ascribed to the electrocatalytic activity of Pd, while the second one originated from the bare GCE. Pd^2+^/COFs/GCE showed three oxidation peaks at −0.50, −0.33, and 0 V (Figure 4c), and no oxidation peak was found at about 0.75 V. This indicates that Pd^2+^/COFs/GCE demonstrated the lowest oxidation over-potential among these modified electrodes, suggesting that the combination of Pd^2+^ and COFs formed new catalytic sites and thus enhanced the oxidation of hydrazine. The different electrodes were also investigated by EIS (Appendix A). The electron transfer rate of different electrodes follows the order: Bare GCE > Pd^2+^/COFs/GCE > COFs/GCE. The higher charge transfer resistance of COFs/GCE is attributed to the lower electrical conductivity.

#### 3.1.3. Analytical Performances for Hydrazine

Figure 5 shows the amperometric current–time response of Pd^2+^/COFs/GCE to the successive additions of hydrazine in 0.1 M NaOH. Upon each successive addition of different concentrations of hydrazine, a stair-shaped slot was raised rapidly and then reached a steady-state oxidation current within 2 s. The inset (a) of Figure 5 shows a linear response over a concentration range from 0.5 μM to 1.6 mM with a sensitivity of 10 μA μM^−1^ cm^−2^ and a detection limit of 0.2 μM (S/N = 3). Appendix A summarizes the sensing performances of hydrazine with Pd^2+^/COFs/GCE and other modified electrodes presented in the literature. The results show that Pd^2+^/COFs/GCE demonstrated an excellent analytical performance for hydrazine, owing to the high electrocatalytic activity.

In order to explore the selectivity of Pd^2+^/COFs/GCE, the interference test was also examined at Pd^2+^/COFs/GCE with 50.0 μM hydrazine in 0.1 M NaOH containing 0.5 mM of the possible interference substances such as KCl, NaCl, Na_2_SO_4_, NaBr, CaCl_2_, NaAc, glucose, (NH_4_)_2_SO_4_, NaNO_3_, and NaNO_2_ (Appendix A). It was found that 100-fold interferents almost did not interfere with the determination of hydrazine. Moreover, a remarkable current response with the addition of hydrazine was observed again after adding the mentioned interference species. The results indicate that our proposed Pd^2+^/COFs/GCE sensor displays excellent selectivity towards the electrochemical oxidation of hydrazine.

The long-term stability of the sensor was further tested by a continuous operation. After 1000s of continuous i-t running for 10.0 μM hydrazine, the electrode still retained 89% of its initial value. In addition, ten successive analyses of 10.0 μM hydrazine using the same electrode showed a relative standard deviation (RSD) value of 4.5%. A reproducible experiment was also evaluated by using five parallel electrodes for the same solution, and the RSD of the current response was found to be 6.4%, suggesting an acceptable reproducibility.

#### 3.1.4. Sample Analysis of Hydrazine

To confirm the practical applicability of the proposed sensor, the validity of Pd^2+^/COFs/GCE for the determination of hydrazine spiked in real samples was tested by a standard addition method (Appendix A). The prepared sensor presented acceptable recoveries of hydrazine in lake water samples (Appendix A).

### 3.2. Electrochemical Sensing of ONP and PNP

#### 3.2.1. Electrochemical Behaviors of ONP and PNP

The electrochemical behaviors of ONP and PNP were investigated at bare GCE (a), COFs/GCE (b), and Pd^2+^/COFs/GCE (c) by CV at a scan rate of 50 mV s^−1^ in 0.1 M PBS (pH = 7.0) (Figure 6). The CVs of these electrodes in 0.1 M PBS are shown in Appendix A. For PNP, three redox peaks were observed, with one main irreversible cathodic peak located at −0.79 V (R_1_) and a pair of reversible redox peaks centered at about 0.08 V (R_2_) and 0.14 V (O_1_) at bare GCE and COFs/GCE. Pd^2+^/COFs/GCE also exhibited the same redox peaks, but the peak of R_1_ shifted to −0.56 V. However, at bare GCE and COFs/GCE, upon the addition of ONP, four redox peaks, including the three peaks and an additional oxidation peak (O_2_) appeared at −0.68 (R_1_), −0.30 (R_2_), −0.25 (O_1_), and −0.23 V (O_2_), respectively. Pd^2+^/COFs/GCE demonstrated the same redox peaks but a much lower R_1_ peak (−0.54 V). This indicated that Pd^2+^/COFs/GCE presented superior electrocatalytic activity towards the R_1_ reduction process of ONP and PNP, which was ascribed to the combination of the high catalytic performance of Pd^2+^ and efficient supporting effect of COFs.

When the potential was scanned negatively, both ONP and PNP could be reduced to hydroxylaminophenol through a four-electron and four-proton transfer process (R_1_). Reversible transformation (O_1_/R_2_) between hydroxylaminophenol and nitrosophenol then occurred with the potential scanning back. In addition, the phenolic group of ONP gave further oxidation through a proton and an electron transfer process (O_2_). The detailed reaction mechanism was proposed as Figure 1 [18]:

To further investigate the electrochemical performance of Pd^2+^/COFs/GCE, the effect of the scan rate on the redox peak current (*I*_pc_) was studied (Appendix A). The results showed that the peak currents of ONP and PNP were quadratic functions of the square root of the scan rate (ν^1/2^), which suggested that the reactions of ONP and PNP on the surface of Pd^2+^/COFs/GCE were mixed controlled processes (surface-controlled at lower scan rates but diffusion-controlled at higher scan rates).

Next, the effect of the pH on the cathodic peak current and potential (R_1_) of ONP and PNP (500 μM) was studied by DPV on the Pd^2+^/COFs/GCE in 0.1 M PBS (Appendix A). The reduction peak current of both ONP and PNP increased firstly and then decreased and reached the maximum at pH = 7.0. In the basic solution, the protonation of PNP and ONP decreases, and the electron density of the nitro group increases due to the decrease in proton concentration, thus hampering the electrochemical reduction of PNP and ONP [19]. Therefore, pH 7.0 was selected for further electrochemical experiments.

#### 3.2.2. Analytical Performances for ONP and PNP

Under optimal conditions, the DPV response of Pd^2+^/COFs/GCE for different concentrations of ONP and PNP in 0.1 M PBS (pH = 7.0) was investigated, respectively (Figure 7a,b). The reduction peak current increased gradually with the concentration of ONP and PNP and exhibited a log–log linear relationship in the range of 5.0 μM–2.0 mM. The detection limit of ONP and PNP was estimated to be 1.6 and 1.3 μM (S/N = 3), respectively. Pd^2+^/COFs/GCE was also applied for the simultaneous detection of ONP and PNP. Figure 7c clearly shows the well-separated reduction peak for ONP and PNP. The calibration curves reveal the favorable linear relationships between the logarithm of peak current and the logarithm of concentrations (5.0 μM–2.0 mM), with 1.8 and 0.9 μM for ONP and PNP, respectively. The electrochemical sensing performances based on Pd^2+^/COFs/GCE and other reported sensors are summarized in Appendix A. The results show that the Pd^2+^/COFs/GCE sensor exhibits comparable performances with the reported methods for the detection of ONP and PNP.

The antifouling property of different electrodes was investigated by successive CVs in the ONP and PNP solution, respectively (Appendix A). The results showed that COFs/GCE exhibited the best antifouling property because of the hydrophilicity of COFs. For ONP, Pd^2+^/COFs/GCE showed improved antifouling properties than GCE in the first six cycles. However, for PNP, both Pd^2+^/COFs/GCE and bare GCE exhibited lower antifouling properties.

### 3.3. Electrochemical Sensing of GSH

#### 3.3.1. Characterization of PC

The morphology of the PCs was characterized by SEM and TEM images. The SEM image of PCs shows the overlapping structure of stratified lamellae (Appendix A). As shown in Appendix A, the flake morphology can be observed in the TEM image of PCs. Furthermore, XPS characterization was employed to investigate the elemental compositions and chemical states of PCs. Figure 8 shows the XPS spectra of PCs containing C, N, and O elements. The C 1s spectra suggest the coexistence of C-C (283.60 eV), C=C (284.50 eV), C=O (287.70 eV), and π-π (292.30 eV) in PCs (Figure 8a) [20]. The N1s XPS spectra show two characteristic peaks at 397.10 and 400.00 eV (Figure 8b), which are attributed to the C=N and C-N groups, respectively [10]. Figure 8c displays the O 1s XPS spectrum of PCs, including two peaks at 531.30 and 532.20 eV, representing the C=O and C-O bonds, respectively [21]. The XPS spectra suggest that a large number of functional groups are present on the surface of the PCs.

#### 3.3.2. Electrochemical Oxidation of GSH

GSH is an important antioxidant in protecting cells from oxidative damage by oxidizing to form glutathione disulfifide (GSSG), and the balance of GSH and GSSG acts as an indicator of many human illnesses such as several cancers, Parkinson’s disease, Alzheimer’s, and HIV [22]. Metal and non-metal electrocatalysts have been explored for the non-enzymatic detection of biological thiols [23,24,25]. Non-metal electrocatalysts consist of redox-active organic molecules [23,26] and carbon materials, including carbon nanotube [27], ordered mesoporous carbon [28], graphene oxide [29], graphene oxide nanoribbons [30], and derived carbon [31]. In fact, the oxo-functionalities present on these carbons contribute to the electrocatalytic activity towards thiols. Recently, nitrogen-doped carbon has proven effective in not only improving the electrical conductivity but also promoting defect sites in the carbon network for better catalytic activity towards GSH [32]. Here, the N-doped PCs with abundant oxo-functionalities were explored for the electro-oxidation of GSH. Figure 9 shows the CV response for the electrochemical oxidation of GSH at PCs/CPE in 0.1 M PBS (pH 7.0) at a scan rate of 50 mV s^−1^. As can be seen, in the absence of GSH, no anodic peak was observed on the PCs/CPE. However, in the presence of GSH, it could be noticed that a remarkable anodic current emerged at about 0.10 V, and the anodic peak was observed at 0.25 V, which could be ascribed to some oxygen-containing functional groups on PCs promoting the oxidation of GSH [26]. Furthermore, there was a very small anodic peak at 0.28 V, and this could be attributed to the edge plane-defective sites on PCs [33]. Furthermore, no cathodic current peak appeared in the CV curves, confirming that the GSH oxidation was a fully irreversible process. The results suggest that the N-doping and the oxo-functionalities present on PCs enhance the rate of electron transfer and exhibit efficient electrocatalytic activity for the oxidation of GSH.

The pH value is a crucial factor for the electrochemical oxidation of GSH, which was examined in 0.1 M PBS with various pH values containing 5.0 mM GSH on the PCs/CPE (Appendix A). The anodic peak current of GSH increased firstly and then decreased as the solution pH value increased from 3.0 to 5.0 and reached a maximum at pH = 4.0. Accordingly, pH = 4.0 is selected as the optimal condition for further detection. GSH is an important tripeptide consisting of L-cysteine, L-glutamine, and glycine with pKa1 = 2.12 (carboxylic acid of glutaminic acid), pKa2 = 3.59 (carboxylic acid of glycine), pKa3 = 8.75 (thiol of cysteine), and pKa4 = 9.65 (ammonium of glutaminic acid) [34]. Gilbertson et al. investigated the interaction and catalytic oxidation mechanisms between GO and GSH. It was found that the synergism between the adjacent epoxide and hydroxyl groups on the GO surface contributed to the catalytic oxidation of GSH [35]. It was speculated that the pH value of the solution is related to both the activity of oxo-functionalities and the ionization of GSH.

#### 3.3.3. Analytical Performances for GSH

Under the optimal conditions, the amperometric responses of the PCs/CPE were conducted by successive injections of various concentrations of GSH into the stirred 0.1 M PBS (pH = 4.0) solution (Figure 10). Upon each addition of 200 μM GSH, a well stair-shaped amperometric response was raised quickly and reached a steady-state oxidation current within 5 s. The inset (a) of Figure 10 shows the response current of the PCs/CPE to various concentrations of GSH, and it is clear that the current increased rapidly with the addition of GSH. The enhancement of response current is ascribed to the excellent electrocatalytic property of the PCs/CPE. Figure 10b displays the linear correlation between the GSH oxidation current and concentration ranging from 5.0 μM to 1.2 mM and 1.2 to 6.4 mM, respectively. The limit of detection was calculated to be 2 μM (S/N = 3). Nevertheless, as the GSH concentration increased gradually, the sensitivity of the prepared sensor slightly decreased owing to the binding of the sulfur moiety to the electrode surface caused by the oxidation of GSH [26].

The current responses of the PCs/CPE to various concentrations of GSH in the static 0.1 M PBS (pH = 4.0) were also conducted using the amperometric technique. The inset of Appendix A shows that the amperometric current increased linearly with the concentrations of GSH in a wide range from 5.0 μM to 10.0 mM. The limit of detection was also calculated to be 1.0 μM (S/N = 3). Appendix A summarizes the electrochemical sensing performances of the PCs/CPE and other reported sensors. The results indicate that the PCs/CPE exhibits excellent performances compared with other reported electrodes for the detection of GSH.

#### 3.3.4. Sample Analysis of GSH

To further assess the practical applications of the proposed PCs/CPE biosensor, the validity of PCs/CPE for the determination of GSH in spiked ample was investigated by both successive and non-successive addition methods (Appendix A). The recoveries of GSH ranged from 92% to 99% and 92% to 107%, respectively (Appendix A).

## 4. Conclusions

In this work, COFs TpPa-1 was explored as an efficient supporting material and precursor to construct electrocatalytic interfaces for electrochemical sensing, and environmental pollutants and biological thiol were sensitively detected. COFs TpPa-1 possesses abundant oxo-functionalities and nitrogen elements, which endow good water dispersibility, powerful adsorption ability, and enhanced electrical conductivity. The results show that COFs TpPa-1 will be a potential material in synthesizing functional materials for electrocatalysis and electroanalysis.

## Data Availability

The authors confirm that the data supporting the findings of this study are available within the article [and/or] its Appendix A.

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
