# Peer review of "Covalent Organic Frameworks-TpPa-1 as an Emerging Platform for Electrochemical Sensing"

_nanomaterials, 2022, doi:10.3390/nano12172953_

Round 1

Reviewer 1 Report

         The authors synthesized a COF incorporated with Pd(II) for electrochemical sensors toward various analytes. The COF was also converted into porous nitrogen-doped carbon and used for electrochemical GSH sensor. The idea is interesting and novel. Lots of electrochemical experiments and sensing analysis have been conducted to support their claims in electrochemistry and sensing parts. However, the material characterizations are quite insufficient, which makes the materials part less promising. As this is a Journal in the field of Nanomaterials rather than an Electrochemistry Journal, I do not think this is acceptable. Thus, before the acceptance of this work, a major revision is required for the authors to address the following comments.

1.      The material characterizations of Pd2+/COFs are very insufficient. Is the COF still crystalline after the incorporation of Pd? How is the change in the porosity of the COF? What is the Pd loading? The authors should prepare the powder material of Pd2+/COF by treating the COF powder with the Pd solution, and show the XRD patterns and nitrogen adsorption-desorption data of both the COF and the Pd2+/COF. The Pd loading should also be quantified by either EDS or ICP.

2.      The porosity of the COF-derived nitrogen-doped carbon should be reported as well.

3.      The electrical conductivity of the COF and the COF-derived nitrogen-doped carbon should be measured and reported.

4.      NaOH solution was used for hydrazine sensing, and PBS was used for ONP and PNP sensing. Is the COF stable in these solutions? The authors are suggested to report the XRD data of the COF after soaking it in PBS or NaOH.

5.      For Figure 5, it is recommended to compare the CV curves of these electrodes before and after adding ONP and PNP.

6.      Minor comments:

(1)   How to prepare Pd2+/GCE? It’s missing in the experimental part.

(2) The authors stated that COFs/GCE was dipped in 10 mM PdCl2 solution for 30. The unit is missing here. 30 seconds? 30 minutes? 30 hours?

Author Response

Thank you very much for your valuable, thoughtful and instructive comments. We are now submitting the revised manuscript for your further consideration. Please see our point-by-point responses to all the comments attached, and the corresponding revisions in the body of manuscript, were marked in red.

Reviewer 2 Report

Comments:

Manuscript ID: nanomaterials-1873146

In this work, the author reported the “Covalent-organic Frameworks-TpPa-1 as an Emerging Platform for Electrochemical Sensing”. The finding of this study will help to determine the most efficient for both hydrazine oxidation reaction and nitrophenol reduction reaction. However, there are major flaws in the manuscript. I recommend its reconsider after major revision as follows:

1)    The introduction is well structured in a good manner, and most of the things are well supported by the topic and the objectives of the study. However, the author advised describing in detail the innovation and advances in relation to the previous literature and more details on sample preparation.

2)    Authors did not discuss anything about fouling. I am expecting that there would be electrode fouling while analyzing these analytes.

3)    In all the i-t curves, why so strong noisy signals are observed?

4)    In Figure 4, the amperometric response after the 100 µM, the steady–state current is not stable. However, the calibration plot shows linearly. Similarly, Figure 9 amperometric response. The author should explain this and also about stability of this proposed sensor.

5)    Amperometric method parameters should be included in the manuscript.

6)    The use of subjects such as "we, they" should be avoided, and the passive voice should be preferred as much as possible.

7)    Spaces need to be used correctly; typo mistakes in the use of unit format; the formula (including upper and lower indices) should be more formal and keep a consistent number of decimal places.

8)    The authors should calculate the electrochemically active surface area of electrodes by using Ferro/ferricyanide redox probe according to the “Randles−Sevcik equation”.

9)    The use of subjects such as "we, they" should be avoided, and the passive voice should be preferred as much as possible.

10) Figure S-2, why do the interference species with similar molecular structures show not big a response? What are the reasons behind this?

11) The main strength of the current study, which is highly reactive performance and better sensitivity is not clearly written to elucidate the major contribution. The enhanced properties of this compound should be emphasized in terms of surface reactivity and enhancement to show the advancement.

12) Precision and accuracy measurements like repeatability and reproducibility are more important to sensor. The author should perform such an important measurement.

Author Response

(The authors gave the same response as above.)

Round 2

Reviewer 1 Report

The authors have revised the manuscript according to previous comments, thus the manuscript can be accepted for publication.

Reviewer 2 Report

The authors have performed the revision in a perfect manner. So, I strongly recommend publication in nanomaterials.